# Effect of a Flavonoid Combination of Apigenin and Epigallocatechin-3-Gallate on Alleviating Intestinal Inflammation in Experimental Colitis Models

**DOI:** 10.3390/ijms242216031

**Published:** 2023-11-07

**Authors:** Mingrui Li, Benno Weigmann

**Affiliations:** 1Department of Medicine 1, Kussmaul Campus for Medical Research, University of Erlangen-Nürnberg, 91052 Erlangen, Germany; li.mingrui@extern.uk.de-erlangen.de; 2Medical Immunology Campus Erlangen, Friedrich-Alexander University Erlangen-Nürnberg, 91052 Erlangen, Germany

**Keywords:** inflammatory bowel disease, flavonoids, synergy

## Abstract

Inflammatory bowel disease (IBD) is an autoimmune disease that leads to severe bowel symptoms and complications. Currently, there is no effective treatment, and the exact cause of IBD remains unclear. In the last decades, numerous studies have confirmed that flavonoids can have a positive impact on the treatment of IBD. Therefore, this study investigated the protective effect of a flavonoid combination of apigenin and epigallocatechin-3-gallate (EGCG) on IBD. In vitro studies in which Caco-2 cell monolayers were incubated with different concentrations of flavonoids found that the flavonoid-treated group exhibited increased transepithelial electrical resistance (TEER) at high concentrations, indicating a protective effect on the barrier function of the intestinal epithelium. In vivo studies showed that flavonoids significantly attenuated inflammatory levels in both chronic and acute hapten-mediated experimental colitis models in a time- and dose-dependent manner. In addition, the activity of myeloperoxidase (MPO) and the level of proinflammatory cytokines in the colon tissue were significantly reduced. Interestingly, the levels of anti-inflammatory cytokines were also dramatically increased. Finally, flavonoids were found to positively modulate the composition of the gut microbiota in the colon. Therefore, a combination of flavonoids could be a promising therapeutic agent for the future adjunctive treatment of IBD.

## 1. Introduction

Inflammatory bowel disease (IBD), including ulcerative colitis (UC), Crohn’s disease (CD), as well as IBD unclassified (IBDU), is a category of chronic or recurrent inflammatory conditions principally targeting the small and large intestine [1]. Among them, UC is characterized by continuous inflammation mainly located in the colon, whereas CD presents as regional inflammation all over the whole gastrointestinal tract. Moreover, inflammation in UC is confined to the intestinal mucosa, whereas CD involves multiple layers of the intestinal wall [2]. Currently, the prevalence of IBD is increasing worldwide. It is estimated that the number of people suffering from IBD in the Western world could exceed ten million in 2030 [3]. Recent research has indicated that environmental factors, gut microbiota, genetic susceptibility, and excess immunity participate in the pathogenesis of IBD. However, the exact cause of IBD remains uncertain [4]. Based on this, IBD is not possible to cure so far. Therefore, the primary goal of IBD treatment is to ameliorate the inflammation, resulting in the signs and symptoms of patients. In general, IBD treatment involves drug therapy and surgery. The drug treatment of IBD mainly includes antibiotics, anti-inflammatory drugs, immune system suppressors, and biologics, which have troublesome side effects when used for a long time. Moreover, surgery does not cure CD, and the benefits are usually temporary [5]. Recently, alternative medicine has increasingly emerged as a research focus in IBD treatment. This includes the investigation of phytochemicals that offer greater tolerance and fewer adverse effects, which has garnered significant interest among medical scientists.

A recent survey showed the total daily amount of flavonoids in the average Western diet ranges from 200–1000 mg/day and drops to 144 mg/day in IBD patients. In addition, a positive correlation has been established between low dietary flavonoid consumption and severe IBD [6]. A randomized trial indicated that adherence to the Mediterranean diet, which is rich in flavonoids, improved clinical scores and inflammatory markers in children and adolescents with mildly to moderately active IBD [7]. The above trial suggests that the supplementation of flavonoids benefits IBD patients. Moreover, certain flavonoids have potential protective effects against IBD. For instance, Fu et al. utilized a dextran sodium sulfate (DSS)-induced colitis mouse model to evaluate the effect of apigenin on UC and found that apigenin effectively inhibited inflammation and protected the gut barrier via balancing the gut microbiome [8]. Wu et al. found that oral epigallocatechin-3-gallate (EGCG) ameliorated colonic inflammation in a gut microbiota-dependent manner [9]. Chaen et al. found that naringenin promoted recovery from colonic damage via suppression of epithelial tumor necrosis factor-α production and induction of M_2_-type macrophages in colitic mice [10]. In addition, quercetin was reported to be able to alleviate UC via activating aryl hydrocarbon receptors to improve intestinal barrier integrity [11]. Recent studies suggested that a combination of various flavonoids may exhibit synergistic anti-inflammatory effects in IBD [12]. However, only limited research has thus far focused on the synergistic anti-inflammatory properties of a flavonoid combo in IBD.

Flavo-Natin is a combined formulation, mainly containing apigenin and EGCG, with smaller amounts of quercetin and epicatechin. In recent years, Flavo-Natin has found clinical application in the management of diarrhea associated with irritable bowel syndrome (IBS) and in the amelioration of dyspeptic symptoms [13,14]. On the one hand, the symptoms and pathogenesis of IBS are similar to those of IBD. On the other hand, the main components in Flavo-Natin are apigenin and EGCG, which have been confirmed to have the potential to treat IBD. Therefore, we have reason to believe that Flavo-Natin is a suitable candidate for the adjunctive treatment of IBD.

In this study, to evaluate the effect of Flavo-Natin administration on UC, we used a hapten-mediated (oxazolone) experimental colitis model to induce acute and chronic colitis in mice. The histological features of acute colitis, such as epithelial damage and infiltration of neutrophils, macrophages, and lymphocytes in mucosa and submucosa, together with the inflammation of the distal colon and elevated levels of type 2 cytokines by lamina propria T cells, are similar to the features of human UC [15]. Additionally, the oxazolone colitis model may be useful in the study of the inflammatory response of chronic UC since chronic colitis can be induced by repetitive administration of oxazolone [15]. Furthermore, we explored whether Flavo-Natin could alleviate colitis by improving the epithelial barrier integrity, decreasing the impairment of colonic tissue, inhibiting the inflammatory level, and balancing the gut microbiome structure in oxazolone-induced colitis mice.

## 2. Results

### 2.1. Flavonoids Can Enhance the Epithelial Barrier Integrity

Some research studies have reported that quercetin and naringenin dose-dependently enhance the transepithelial electrical resistance (TEER) values of Caco-2 cell monolayers over a range of 10–100 μM [16,17]. In addition, a luminal flavonoid concentration of 100 μM can be attained with a human intake of 50 mg [18]. Therefore, we chose 100 μM of flavonoid (EGCG, naringenin, quercetin) to incubate Caco-2 cell monolayers (see Figure 1C). As for apigenin, we decreased its concentration to 50 μM in order to control dimethyl sulfoxide (DMSO) concentration below 1%. Naringenin and quercetin were also dissolved in DMSO, except for EGCG dissolved in Dulbecco´s Phosphate Buffered Saline (PBS). The concentrations of DMSO were all below 1% in the medium. The impact of distinct flavonoids on the TEER of Caco-2 cell monolayers varied. As shown in Figure 1A, apigenin showed a negative effect on TEER, signifying an augmentation in the intestinal permeability. The TEER values of Caco-2 cell monolayers incubated with apigenin (50 μM) were lower than the control values (0 μM of apigenin) after 1 h, which had statistical significance (*p* < 0.05) at 12, 24 h. Contrastingly, EGCG, naringenin, and quercetin exhibited a positive effect on TEER, indicating a reduction in intestinal permeability. The TEER values for cells exposed to EGCG surpassed the control values after 12 h and had statistical significance (*p* < 0.05) at 24, 36, and 60 h. In addition, naringenin significantly increased the TEER values after 1 h (*p* < 0.05), and quercetin also enhanced the TEER values at each time point (*p* < 0.05).

### 2.2. Flavonoids Can Enhance TEER

To evaluate the effect of Flavo-Natin on the epithelial barrier integrity, we incubated epithelial cell line Caco-2 cell monolayers with 0 μg/mL, 150 μg/mL, and 1 mg/mL Flavo-Natin followed by measuring the TEER value of each Transwell insert. Moreover, DMSO was used to dissolve Flavo-Natin, and its concentration is less than 1% in the complete RPMI 1640 medium to avoid damaging cells. As depicted in Figure 1B, the TEER in monolayers incubated with 1 mg/mL Flavo-Natin was significantly higher than in those incubated without Flavo-Natin (control group) at 36 h and 60 h (*p* < 0.05). However, we did not observe a significant difference in TEER between monolayers incubated with 150 μg/mL Flavo-Natin and the control group. Furthermore, we observed a time-dependent increase in the high-dose group (1 mg/mL Flavo-Natin), with the peak value occurring at 60 h.

To determine the most effective incubation concentration of Flavo-Natin in TEER measurement, a scratch assay was conducted in this study. Based on Figure 2, it was not observed that there was a significant difference in the migration rate of Caco-2 cells between 10 μg/mL Flavo-Natin and without Flavo-Natin (control group). However, the migration rate exposed to 100 µg/mL Flavo-Natin was significantly lower than the control group at 48 h and 72 h (*p* < 0.01). The results suggest that Caco-2 cells can be influenced only when the concentration of Flavo-Natin is higher than 100 µg/mL. Consequently, we chose 150 μg/mL and 1 mg/mL Flavo-Natin to incubate Caco-2 cell monolayers in the above cell experiment.

### 2.3. Flavo-Natin Showed a Time-Dependent Protective Effect on Chronic Experimental Colitis Model

To investigate the protective effect in vivo, chronic oxazolone-induced colitis models were used in mice, and then different doses of Flavo-Natin were administered orally to the colitis mice (Figure 3G). As shown in Figure 3A, we observed that Flavo-Natin, including low-doses and high-doses, did not significantly relieve the inflammatory level of colitis mice after the 1st and 2nd challenges, respectively. However, the murine endoscopic index of colitis severity (MEICS) of the low-dose and high-dose groups was significantly lower than in the placebo group after the 3rd challenge. In addition, there was no statistical difference in the MEICS score between the low-dose and high-dose groups after the 3rd challenge (*p* > 0.05). To confirm the conclusion, we made a pathological section of colon tissue and found that the histological differences between the three groups were consistent with the MEICS differences after the 3rd challenge (Figure 3B). Furthermore, we made in vivo imaging of myeloperoxidase (MPO) activity via an in vivo imaging system (IVIS) after the 3rd challenge, then found that the low-dose and high-dose groups expressed a lower luminescence and accordingly less active MPO compared with the placebo group (Figure 3D). This result was subsequently corroborated through immunofluorescence staining of MPO on cryosections. The placebo group exhibited a higher number of MPO-positive cells. Contrastingly, the low-dose and high-dose groups displayed a significantly reduced quantity of MPO-positive cells (Figure 3C,E). Moreover, the difference between the low-dose and high-dose groups was not statistically significant.

Subsequently, we performed quantitative polymerase chain reaction (qPCR) to analyze if Flavo-Natin affected the expressions of proinflammatory and anti-inflammatory cytokines. The relative expressions of interleukin (*Il*)*1β*, *Il6*, tumor necrosis factor (*Tnf* )-*α*, and *Il*10 to the housekeeping gene *18s rRNA* were calculated. Based on the above endoscopic and histological analyses, we only isolated mRNA from the colon tissue after the 3rd challenge. As shown in Figure 3F, mice treated with Flavo-Natin showed lower expressions of *Il1β, Il6, Tnf-α* at the mRNA level compared with placebo mice (*p* < 0.01). Additionally, the high-dose group showed lower expressions of these proinflammatory cytokines compared with the low-dose group (*p* < 0.05). Moreover, the low-dose and high-dose groups showed a higher expression of *Il1*0 than the placebo group. Additionally, the high-dose group expressed a higher level of *Il1*0 than the low-dose group. To summarize, a protective effect of Flavo-Natin was observed in the chronic inflammatory model after the third challenge.

For the analysis of flavonoids regulating intestinal microbiota, we collected the stools after the starting point and 2 days after the 1st, 2nd, and 3rd challenges, respectively. Deoxyribonucleic acid (DNA) was isolated from the stools and analyzed. As shown in Figure 4, we observed that the proportion of *Alloprevotella* in the low-dose and high-dose groups was higher than that in the placebo group after the starting point. However, the proportion of *Alloprevotella* in treatment groups gradually decreased after the 1st, 2nd, and 3rd challenges (cycles 1, 2, 3). Moreover, the proportion of *Akkermansia* in both the low-dose and high-dose groups was lower than in the placebo group in cycle 0. In contrast, in cycles 1, 2, and 3, the proportion of *Akkermansia* was higher in the treatment groups than in the placebo group, supporting the protective effect of Flavo-Natin on a positively assessed microbiota.

### 2.4. Flavo-Natin Showed a Protective Effect on Acute Oxazolone-Induced Colitis, Which Is Dose-Dependent

In addition to chronic colitis models, we also utilized acute colitis models to evaluate the impact of Flavo-Natin on IBD (Figure 5G). Data showed that the high-dose group obtained a lower MEICS than the placebo group (*p* < 0.01), which did not occur in the low-dose group (Figure 5A). Furthermore, histological comparison indicated that 3 mg/day of Flavo-Natin significantly relieved the inflammatory level of colitis models compared with the placebo group (*p* < 0.001). However, 1 mg/day of Flavo-Natin did not obtain a lower histological score than the placebo group (Figure 5B). Moreover, the low-dose and high-dose groups displayed a lower luminescence and, accordingly, less active MPO than the placebo group (Figure 5C). However, significantly reduced MPO-positive cell counts were observed exclusively in the high-dose group, as shown in Figure 5D (*p* < 0.001). From Figure 5E, the high-dose group showed lower expressions of *Il1β, Il6, Tnf-α* at the mRNA level compared with the placebo group, which was consistent with those in chronic colitis models. However, the low-dose group did not show lower expressions of these proinflammatory cytokines than the placebo group. Additionally, both the low-dose and high-dose groups showed a higher expression of *Il10* than the placebo group. In conclusion, we observed that the above regulatory effect on cytokines is dose-dependent. Furthermore, cytokine expressions in the plasma were analyzed via enzyme-linked immunosorbent assay (ELISA). We measured the concentrations of IL1β, IL6, TNF-α, and IL10 in plasma and found that the expression levels of TNF-α and IL10 in plasma were too low to be detected. However, the plasma concentrations of IL1β and IL6 in the high-dose and low-dose groups were significantly lower than in the placebo group. Unlike the qPCR results, the results of ELISA indicated that low-dose Flavo-Natin likewise significantly reduced the expression levels of proinflammatory cytokines (IL1β, IL6) compared with the placebo group (Figure 5F).

## 3. Discussion

Numerous studies have confirmed that certain flavonoids have a considerable prospect for treating IBD [19,20,21,22,23]. To date, pathways have been found to participate in the regulation of IBD mediated by flavonoids, in which the preservation of the intestinal epithelial barrier is a key step in preventing the aggravation of IBD [24]. Flavonoids have previously been demonstrated to improve the epithelial barrier. For instance, Watson et al. found that EGCG blocked epithelial barrier dysfunction induced by interferon-γ (*IFN-γ*) by measuring TEER in T84 (human colonic epithelial cell) monolayers [25]. In addition, quercetin and naringenin have been found to dose-dependently increase the TEER of Caco-2 cell monolayers within the concentration range of 10–100 µM [16,17]. Consistent with these findings, EGCG, quercetin, and naringenin significantly increased the TEER of Caco-2 cell monolayers in our study, indicating the preservation of the intestinal epithelial barrier by flavonoids. However, not all flavonoids possess the capability to enhance the epithelial barrier. In our observations, apigenin significantly decreased the TEER of Caco-2 cell monolayers, suggesting the dysfunction of the intestinal epithelial barrier. Moreover, Flavo-Natin, containing apigenin and EGCG, significantly increased the TEER of Caco-2 cell monolayers at a high concentration. However, no notable difference was observed at a low concentration. The discrepancy may be attributed to apigenin potentially counteracting the positive effect of EGCG. When Caco-2 cell monolayers were incubated with a low concentration of Flavo-Natin, apigenin, a main component of Flavo-Natin, greatly offset the increase in TEER induced by EGCG. As a result, we did not observe an enhancement in TEER values due to the low concentration of Flavo-Natin. However, at a high concentration of Flavo-Natin, the amount of quercetin significantly increased. Furthermore, quercetin demonstrated a substantial increase in TEER values (see Figure 2). Therefore, apigenin did not counteract the augmentation of TEER values induced by both EGCG and quercetin. 

It has been reported that a combination of multiple phytochemicals, including flavonoids, can lead to synergistic anti-inflammatory activity [26]. For instance, the combined application of EGCG and genistein resulted in an improvement in EGCG uptake within the small intestine [27]. Moreover, the combined administration of quercetin and resveratrol exhibited a synergistic anti-inflammatory effect in rats subjected to a high-fat–sucrose diet [28]. Therefore, it is plausible to consider that the concurrent administration of multiple flavonoids could potentially evoke synergistic anti-inflammatory properties in IBD. In this study, Flavo-Natin as a flavonoid combo, at a dose of 0.2 mg/day and 0.8 mg/day, was used to treat the chronic colitis mice induced by oxazolone. However, the reduced inflammation measured by MEICS was not observed in the low-dose and high-dose groups after the 1st and 2nd challenges, compared with the placebo group. Only after the 3rd challenge were significant differences in inflammatory extent (MEICS, histological score, MPO activity, and cytokine levels) between the treatment and placebo groups evident, indicating the time-dependent therapeutic effect of Flavo-Natin. Moreover, we observed alterations in the intestinal microbiota in chronic colitis models, revealing an increase in the proportion of *Alloprevotella* and *Akkermansia* following oral administration of Flavo-Natin. Recent studies have found the crucial role of *Alloprevotella* in maintaining the intestinal epithelial barrier and improving intestinal immune function [29], while *Akkermansia* is able to regulate the thickness of the intestinal mucosa layer and strengthen therapeutical outcomes in chronic diseases associated with leaky gut and inflammation [30]. Furthermore, in alignment with our results, Wu et al. observed that oral administration of EGCG significantly increased *Akkermansia* abundance in mice, leading to the attenuation of colitis. The potential protective mechanism may involve the substantial enhancement of short-chain fatty acids (SCFAs) production by *Akkermansia*. These SCFAs contribute to an anti-oxidative and anti-inflammatory state, thus providing further protection against colon damage [9]. In addition, apigenin also significantly enhanced the abundance of *Akkermansia* and *Faecalibaculum*. These beneficial bacteria produced a large amount of SCFAs, mainly including acetic acid, propionic acid, and butyric acid. Subsequently, microbiota-derived SCFAs upregulated IL-22 production, which protects intestines from inflammation to maintain intestinal homeostasis [8]. In our study, the protective pathway of alteration in intestinal microbiota has not been investigated. However, considering Flavo-Natin as a combination of apigenin and EGCG, we have grounds to believe that an increase in the proportion of *Alloprevotella* and *Akkermansia* could stimulate the production of SCFAs, which is likely associated with the mitigation of colitis.

The results in chronic colitis models were not as favorable as anticipated. We hypothesized that the failure of Flavo-Natin to alleviate inflammation after the 1st and 2nd challenges may be attributed to an insufficient dosage. Conversely, the success observed after the 3rd challenge can be attributed to the cumulative effect of Flavo-Natin over time. Additionally, Fu et al. demonstrated significant improvement in ulcerative colitis in chronic colitis mice induced by DSS through the administration of apigenin at doses of 3 mg/day and 5 mg/day [8]. Wu et al. effectively alleviated experimental colitis by orally administering 1 mg/day of EGCG to acute DSS-induced colitis mice [9]. It should be noted that the doses of apigenin and EGCG utilized in the above studies were considerably higher than those employed in our study. To validate this hypothesis, higher doses of Flavo-Natin were administered to mice with acute oxazolone-induced colitis. Significant disparities in inflammation were observed between the treatment groups and placebo group in acute colitis models when having a gavage of Flavo-Natin at a dose of 3 mg/day, as assessed by MEICS, histological score, MPO activity, and cytokine levels. These findings suggest a dose-dependent correlation between the administration of Flavo-Natin and its therapeutic efficacy in IBD. Besides, Fu et al. employed oral administration of apigenin to treat chronic colitis in mice induced by DSS and discovered that a daily dosage of 5 mg exhibited a more robust anti-inflammatory effect than a dosage of 3 mg [8]. Xu et al. investigated the therapeutic effect of EGCG on acute colitis in mice induced by DSS, revealing that a dosage of 2 mg/day produced a better therapeutic outcome compared with a dosage of 1 mg/day [31]. These observations align with our findings, suggesting that both Flavo-Natin and its primary components, apigenin, and EGCG, indicate a dose-dependent correlation between their oral administration and the therapeutic effect in IBD, despite variances in experimental setups.

In conclusion, the time- and dose-dependent protective effects of Flavo-Natin on IBD were demonstrated via both chronic and acute colitis models induced by oxazolone, supporting the importance of the protective effect of Flavo-Natin in the inflammatory bowel disease model. However, there are some limitations in our study. First, we only employed an oxazolone-induced colitis model to assess the protective effects of Flavo-Natin on IBD, which can partially mimic IBD to a certain extent but not replicate it fully. In the next animal experiment, we intend to utilize a DSS-induced colitis model to address this limitation comprehensively. Second, the sample size of the collected stool specimens in this study was relatively small. Therefore, the reliability of our intestinal microbiome analysis may focus on the small samples but can be increased in the future. Lastly, we did not conduct a comparative analysis of Flavo-Natin with apigenin and EGCG regarding their therapeutic efficacy in IBD. We plan to address these limitations in our future study.

## 4. Materials and Methods

### 4.1. Cell Culture

Caco-2 cells were obtained from the European Collection of Authenticated Cell Cultures (ECACC) and cultured in complete RPMI 1640 Medium supplemented with 10% Fetal Calf Serum (FCS), 1% GlutaMAX (100×), 1% Sodium Pyruvate (100×), Penicillin (100 U/mL), and Streptomycin (100 µg/mL) at 37 °C in a 5% CO_2_ incubator. In our study, Caco-2 cells were from passage number 20–50.

### 4.2. Transepithelial Electrical Resistance (TEER)

TEER often serves as a rough indicator of tight junction permeability; here, we performed a TEER measurement to evaluate intestinal epithelial barrier integrity in Caco-2 cell monolayer incubated by different drugs. Briefly, Caco-2 cells were seeded at a density of 50,000 cells/mL on Transwell inserts (24-well, 0.4 µm pore size, 0.336 cm^2^ area, transparent, Greiner Bio-One Co., Kremsmünster, Austria) and cultured in complete RPMI 1640 medium for 21 days, which was replaced twice a week. Based on the method of Yamashita et al. [32], TEER values were measured at 20 °C using a Millicell-ERS resistance monitoring apparatus (Millipore, Burlington, MA, USA). The chopstick electrodes, connected to the epithelial volt-ohmmeter, had to be calibrated to 0 ohm (Ω) before each measurement. Subsequently, they were placed into two chambers of each Transwell insert to measure TEER values. The TEER value (in Ω × cm^2^) was calculated by subtracting the background value (see the following equation).
TEER value (Ω × cm^2^) = (Total resistance − blank resistance) (Ω) × Area (cm^2^).

After culturing for 21 days, only Caco-2 cell monolayers with TEER values above 400 Ω × cm^2^ were used for further experiments. Whereafter, the qualified Caco-2 cell monolayers were incubated with different concentrations of drugs for 72 h, which were dissolved in a complete RPMI 1640 medium. At regular intervals, the TEER value of each insert was measured in triplicated and recorded. Over 72 h of drug incubation, the cell medium was not changed.

### 4.3. Immunofluorescence Staining of Caco-2 Cell Monolayer

A total volume of 100 μL of 4.5% Paraformaldehyde (PFA) was used to fix Caco-2 cell monolayers for 10 min. Subsequently, PFA was removed, and PBS (200 μL) was used to wash the residual PFA. The borders of Caco-2 cells were stained with 100 μL of rhodamine-phalloidin (1:50) in buffered PBS + 0.2% Triton-X 100 for 15 min. PBS (200 μL) was used to wash the residual dye after removing the solution, and 100 μL of DAPI (1:5000) was used to stain the cell nucleus for 5 min. Finally, the inserts were cut and mounted on glass slides with Mowiol after removing the DAPI solution. The slides were stored at 4 °C, and the above steps were all performed in the dark. Images were captured using a confocal microscope.

### 4.4. Scratch Assay

Silicone chambers, purchased from the Ibidi company, were utilized for culturing Caco-2 cells. First, 350,000 Caco-2 cells in 70 µL medium were evenly added into each of the two silicone chambers via pipette. These chambers were subsequently transferred into an incubator (37 °C and 5% CO_2_) for 72 h until the gap between the two silicone chambers was colonized by Caco-2 cells. The silicone chambers were then removed, and 2 mL of medium was added to the chambers. This medium contained different concentrations of Flavo-Natin (0 μg/mL, 10 μg/mL, 100 μg/mL). The chambers were analyzed after 24, 48, and 72 h. At different time points, the images were captured under a microscope, and the migrated distance was measured using Leica Application Suite (LAS) software SP5. The migration rate was calculated using the following formula.
Migration rate (%) = (D0 h − D h)/ D h × 100

In the above formula, D0 h represents the distance of the gap at 0 h, and D h represents the distance at 24, 48, and 72 h.

### 4.5. Mice

For our experiments, 8–10-week-old female C57BL/6 mice (weighing 18–20 g) were obtained from Jackson Laboratory. All mice were maintained on a 12-h light-dark cycle at 23 ± 1 °C with a relatively constant humidity of 50 ± 5% under specific pathogen-free (SPF) conditions and had free access to water and standard laboratory pellets. The health and potential infestation with microbes and parasites were regularly analyzed by our veterinarian. Before the experiment, the mice were given 5 days to adapt to our animal facility. The animal study was performed in accordance with institutional guidelines and under the permission of the government of Middle Franconia in Germany.

### 4.6. Acute and Chronic Oxazolone-Induced Colitis Models

For acute oxazolone-induced colitis, the first step was sensitization. In this process, the abdomens of the mice were shaved, followed by a topical administration of 100 μL of 3% oxazolone solution. They were then challenged using oxazolone solution 5 days after sensitization, and the operation was as follows: 1% oxazolone solution (100 μL) was administered into the colon of the mice after they were anesthetized. We used a mini endoscopy to evaluate the severity of colitis 24 h after the challenge. Finally, all mice were euthanized 48 h after the challenge, and their organ tissues were removed to be further analyzed. Compared with acute oxazolone-induced colitis, chronic oxazolone-induced colitis can mimic the characteristic symptoms of UC better. Therefore, we also set up chronic oxazolone-induced colitis models. First, sensitization was conducted. Afterward, the challenges were repeated for 3 cycles at a 0.5% oxazolone concentration, followed by cycles of recovery. The extent of colitis was analyzed with a mini endoscopy 24 h after each challenge. Finally, all mice were euthanized 48 h after the last challenge [15].

### 4.7. Administering Medications to Mice in Colitis Models

Flavo-Natin is a combined formulation, mainly containing apigenin and EGCG, with smaller amounts of quercetin and epicatechin. For the composition of Flavo-Natin, see Table 1. First, different amounts of Flavo-Natin were diluted in 2% carboxymethylcellulose sodium (CMC-Na) solutions. To observe its protective effect on IBD, 100 μL of Flavo-Natin suspension was orally administered in acute or chronic oxazolone-induced colitis mice as treatment groups. Contrastingly, the mice accepted 100 μL of 2% CMC-Na solution were considered as a placebo group. The mice were orally administered Flavo-Natin from the start to the end in acute or chronic oxazolone-induced colitis models.

In regard to the acute oxazolone-induced colitis models, all the mice were randomly distributed over the cages and divided into 3 groups: placebo group, low-dose group, and high-dose group (*n* ≥ 5 per group). We orally administered 100 μL of 2% CMC-Na solutions to the mice in the placebo group each day. In addition, 100 μL of Flavo-Natin suspensions, in amounts of 1 mg and 3 mg, were orally administered in the low-dose and high-dose groups each day, respectively. All mice were induced with colitis with oxazolone (Figure 5G).

As for the chronic oxazolone-induced colitis models, we randomly distributed all mice over the cages and divided them into 3 groups: placebo group, low-dose group, and high-dose group (*n* ≥ 4 per group). For the placebo group, 100 μL of 2% CMC-Na solution was orally administered in mice. Contrastingly, 100 μL of Flavo-Natin suspension, at a concentration of 2 mg/mL or 8 mg/mL, was orally administered in the low-dose group or high-dose group. Three groups were then induced with colitis with oxazolone (Figure 3G).

To minimize observational bias, one researcher was responsible for the randomization process of allocating mice to different groups and oral administration of drugs each day. Contrastingly, another researcher, blinded to the treatment groups, was in charge of establishing acute and chronic oxazolone-induced colitis models and subsequently conducted endoscopic and tissue analyses.

### 4.8. Endoscopic Analysis

A mini endoscopy analysis was conducted to assess the severity of colonic inflammation 24 h after each challenge by oxazolone during the colitis course. First, the mice were anesthetized with 2% isoflurane by placing them in a chamber. The mini endoscope was then introduced rectally, and photographs were captured. The indicators outlined in Table 2 were assessed and graded on a scale of 0–3 for each indicator. These scores were aggregated, and the MEICS was determined accordingly [33]. The MEICS spans from 0 (the absence of inflammation) to 15 (severe inflammation).

### 4.9. Comparison of MPO Activity Using IVIS

In vivo staining to detect MPO activity was performed 24 h after challenge during the oxazolone-induced colitis course. MPO is a key marker for inflammation and is predominantly expressed by neutrophils and macrophages. It mediates the conversion of H_2_O_2_/chloride ions into hypochlorous acid. Luminol interacts with hypochlorous acid, then emitting luminescent light that is detectable. This luminescence emission can offer insights into MPO activity [34]. To measure MPO activity, mice were intraperitoneally injected with 20 nM luminol L-012 (100 µL) and anesthetized using 2% isoflurane in a chamber. With their shaved bellies facing upwards, the mice were placed in the IVIS chamber. Luminescence was then measured for 1 min.

### 4.10. Stool Microbiome Test

Recent studies found that IBD is associated with intestinal dysbiosis [35]. In addition, some flavonoids can modulate gut microbiota to prevent and treat IBD [8]. To explore the correlation between flavonoids and the alteration of intestinal flora, we collected the stool derived from the mice in chronic colitis models. The stools were collected 2 days after the starting point and the following 1st, 2nd, and 3rd challenges. Afterward, total microbial genomic DNA was isolated from each fecal sample. The concentration of DNA was measured by a NanoDrop 1000 device. DNA was stored at −20 °C for further studies. Finally, all DNA samples were tested through 16s rRNA gene sequencing, and microbiota data were analyzed.

### 4.11. Hematoxylin and Eosin (H and E) Staining

The colon tissue was fixed on the slides and stained with hematoxylin and eosin. Afterward, bright field microscopy was used to capture images. All H and E staining slides were analyzed by a pathologist, in which the extent of colitis was scored in Table 3 [15]. 

### 4.12. MPO Staining of Tissue

The colon tissue on the slides was fixed by 4.5% PFA for 15 min. Subsequently, the slides underwent a triple wash with PBS, and the tissue was subjected to permeabilization using 1× Permeabilization buffer in PBS for 10 min. To minimize nonspecific antibody binding, a blocking solution was applied and incubated for 30 min. Following this, the colon tissue on the slide was intracellularly stained for MPO using a goat antibody (dilution 1:50). The antibody was then appropriately diluted in antibody dilution buffer, and the tissue sections were incubated with this antibody solution overnight at 4 °C. Following overnight incubation, the slides were once again washed thrice with PBS-Tween, and a rabbit-anti-goat antibody (dilution 1:200) was applied for 2 h at room temperature. After another round of PBS washing (3 times), the cell nuclei were stained with DAPI (1:5000 in PBS) for 5 min. Finally, a cover slip was affixed using Mowiol. The immunofluorescence images were captured using a confocal microscope. During the process of MPO staining, positive and negative controls were established to demonstrate the reliability of the results. Negative control slides were incubated with PBS instead of primary and secondary antibodies, whereas positive control slides were incubated with PBS and secondary antibodies.

### 4.13. RNA Isolation, Synthesis of cDNA, and Quantitative PCR (qPCR)

Total RNA was isolated from the colonic tissues of mice using the Nucleo Spin RNA Kit. After measuring the concentration of RNA, cDNA was synthesized from 500 ng RNA using the SCRIPT cDNA Synthesis Kit. Synthesized cDNA was stored at −20 °C and diluted with nuclease-free water in a 1:10 ratio before conducting a qPCR assay. The housekeeping gene *18S rRNA*, exhibiting constitutive expression in the inflamed colon tissue, served as the reference gene for the target genes. Table 4 lists the primers of target genes, including *Il1β*, *Il6*, *Tnf-α*, and *Il10*. The mRNA levels of these target genes were calculated by the 2^−△△Ct^ method. All assays were repeated independently in triplicate to enhance the robustness of the results.

### 4.14. Enzyme-Linked Immunosorbent Assay (ELISA)

After the death of the mice, the plasma was immediately separated from blood samples via centrifugation at 12,500 rpm and subsequently stored at −20 °C for further study. The concentrations of IL-1β, IL-6, IL-10, and TNF-α in the plasma were determined using ELISA. In brief, 96-well plates were coated with the respective capture antibody (50 µL) in duplicate, followed by incubation overnight at 4 °C. On the second day, they were washed 3 times using 200 µL of PBS-Tween (0.05%). A total volume of 100 µL of assay diluent was then used to incubate the wells for 1 h at room temperature. Subsequently, samples (50 µL) or the respective standards were incubated in the wells overnight at 4 °C after a rinse with PBS-Tween. It was worth noting that all samples were diluted 1:2 with assay diluent. On the third day, the wells were washed again, and a detection antibody (50 µL) was added to the wells, which were slightly shaken for 1 h at room temperature. Avidin-HRP (50 µL) reagent was added into the wells after a rinse with PBS-Tween. The plates were incubated for 30 min in the dark. Thereafter, 50 µL of substrate solution was added into each well after washing again. The plates were placed in the dark for 15–30 min. Finally, 50 µL of 2N H_2_SO_4_ was added to each well to stop the chemical conversion. The absorbance was measured at 450 nm, simultaneously with a reference wavelength (570 nm), using an ELISA reader instrument.

### 4.15. Statistics

Statistical differences between the two groups were assessed using paired Student’s *t*-tests. When statistical differences between more than two groups were assessed, one-way ANOVA with an additional Tukey post-test was used. Statistical analysis was performed using GraphPad Prism 9.0. *p* values < 0.05 were considered statistically significant and marked with asterisks (* < 0.05, ** < 0.01, *** < 0.001, **** < 0.0001). In the *t*-test for independent samples, the effect size was determined according to Cohen’s by calculating the mean difference between the two groups and dividing the result by the pooled standard deviation. Data were presented as means +/− standard error of the mean (SEM). D-value in this study was qualified.

## 5. Conclusions

Collectively, the observations of this study revealed the roles of Flavo-Natin in IBD treatment. According to our findings, Flavo-Natin exerted protective functions in the oxazolone-induced colitis model, which is time- and dose-dependent. Moreover, the mechanisms of Flavo-Natin treating IBD may be attributed to the maintenance of the intestinal epithelial barrier, antioxidant properties, shaping microbiota, and the modulation of proinflammatory and anti-inflammatory cytokines. 

## Figures and Tables

**Figure 1 ijms-24-16031-f001:**
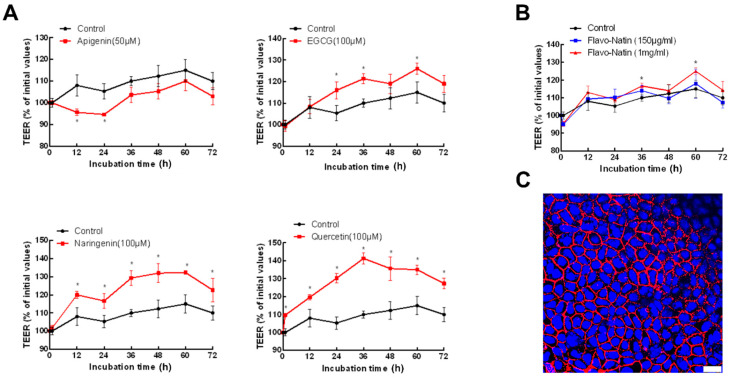
TEER for Caco-2 cell monolayers exposed to apigenin, EGCG, naringenin, quercetin, and Flavo-Natin. TEER is shown as a percent of initial values. (**A**) Caco-2 cell monolayers were incubated with apigenin (50 μM), EGCG (100 μM), naringenin (100 μM), and quercetin (100 μM) for 72 h. (**B**) Caco-2 cell monolayers were incubated with 150 μg/mL and 1 mg/mL Flavo-Natin as well as control for 72 h. (**C**) Immunofluorescence image of Caco-2 cell monolayer in TEER measurement. Cell borders were stained in red with rhodamine-phalloidin, and cell nuclei were stained in blue with DAPI. The pictured scale represents 25 µm. All results are the mean values ± SEM, *n* ≥ 5 per group. * *p* < 0.05 vs. Control group.

**Figure 2 ijms-24-16031-f002:**
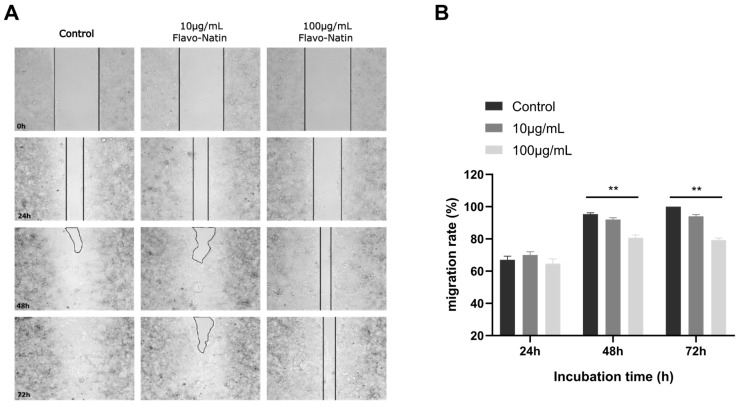
In vitro scratch assay performed to analyze the dose effect of Flavo-Natin on the migration capability of Caco-2 cells. (**A**) Representative images from in vitro scratch assays. The cells were incubated for 72 h with different concentrations of Flavo-Natin. Cell migration was documented via bright field microscopy. (**B**) The comparison of migration rate between different groups at 24 h, 48 h, and 72 h. All results are the mean values ± SEM, *n* ≥ 5 per group. ** *p* < 0.01 vs. Control.

**Figure 3 ijms-24-16031-f003:**
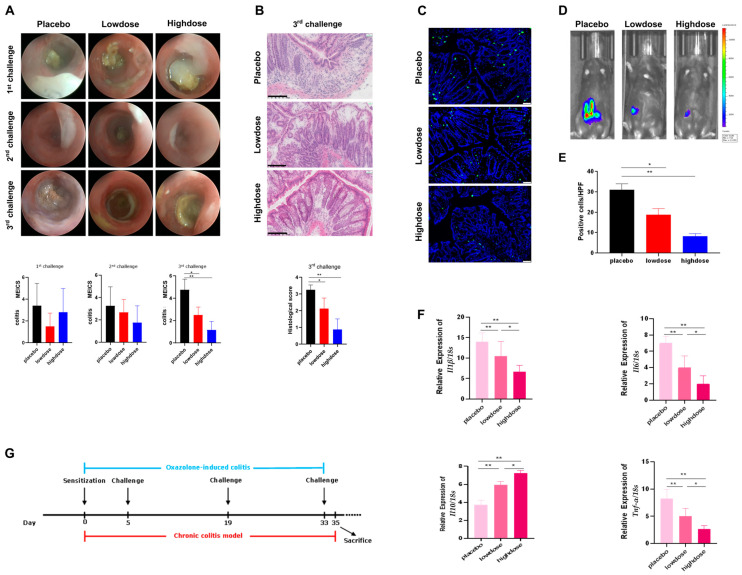
Chronic experimental colitis models treated with Flavo-Natin show reduced signs of inflammation. All results are the mean values ± SEM, *n* ≥ 4 per group. Statistical analyses were performed using ANOVA (* *p* < 0.05, ** *p* < 0.01). (**A**) Mini endoscopic images after each challenge and their corresponding MEICS values. (**B**) Histological pictures of colon tissue and their corresponding scores. The pictured scale represents 100 µm. (**C**) Immunofluorescence images of MPO in colon tissue. The pictured scale represents 50 µm. (**D**) IVIS images of colitis models. (**E**) Quantitative assessment of MPO-positive cell counts. (**F**) Flavo-Natin leads to reduced expression of *Il1β*, *Il6*, *Tnf-α* and increased expression of *Il10* at the mRNA level. (**G**) Schematic overview of the chronic oxazolone colitis models. The mice were orally administered Flavo-Natin or placebo from day 0 to day 35.

**Figure 4 ijms-24-16031-f004:**
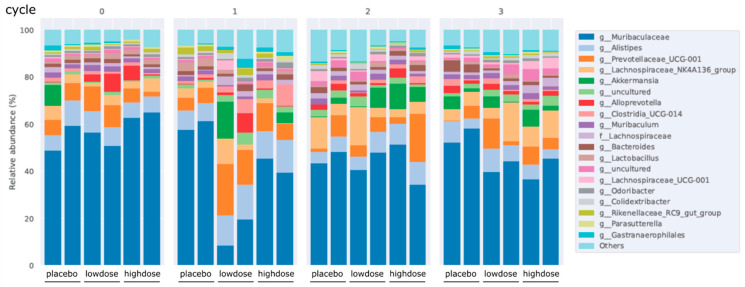
Flavo-Natin regulates the composition of intestinal microbiota. The stools were collected 2 days after the starting point and the following 1st, 2nd, and 3rd challenges, respectively. DNA was then isolated from the stools. These intestinal microbiota data were analyzed via *16s rRNA* gene sequencing.

**Figure 5 ijms-24-16031-f005:**
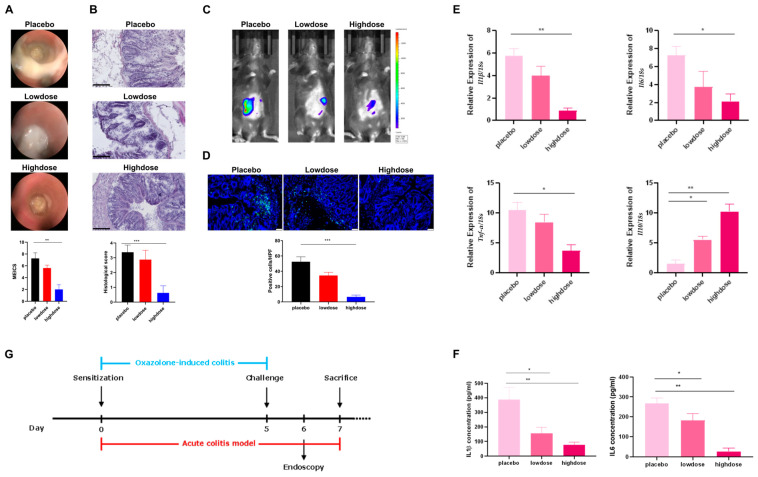
Acute experimental colitis models treated with Flavo-Natin show reduced signs of inflammation. (**A**) Mini endoscopic images after challenge and their corresponding MEICS values. (**B**) Histological pictures of colon tissue and their corresponding scores. The pictured scale represents 100 µm. (**C**) IVIS images of colitis models. (**D**) Immunofluorescence images of MPO in colon tissue. The pictured scale represents 50 µm. Quantitative assessment of MPO-positive cell counts. (**E**) Flavo-Natin leads to decreased expression of *Il1β, Il6, Tnf-α* and increased expression of *Il10* at the mRNA level. (**F**) Flavo-Natin blocks the secretion of proinflammatory cytokines via ELISA. The serum was collected after sacrificing mice; plasma was then isolated via centrifuging. Subsequently, the levels of IL1β, IL6, TNF-α, and IL10 in plasma were analyzed. (**G**) Schematic overview of the acute oxazolone colitis models. The mice were orally administered Flavo-Natin or placebo from day 0 to day 7. All results are the mean values ± SEM, *n* ≥ 5 per group. * *p* < 0.05, ** *p* < 0.01, *** *p* < 0.001 vs. placebo group.

**Table 1 ijms-24-16031-t001:** The component of Flavo-Natin.

Component	Amount
Epigallocatechin gallate	10.4 mg
Apigenin-7-Glucosid	10.0 mg
Epicatechin gallate	4.0 mg
Epigallocatechin	2.1 mg
Epicatechin	1.4 mg
Quercetin	0.6 mg
Catechin	0.3 mg
Myricetin	0.2 mg

Note: One capsule of Flavo-Natin (0.85 g), composed of caffeine 6 mg, vitamin C 55 mg, folic acid 110 μg, vitamin B6 1.3 mg, vitamin B12 2 μg, inulin (23%), and some excipients [13,14].

**Table 2 ijms-24-16031-t002:** Indicator for the assessment of the MEICS.

Indicator	Score
Transparency of the colon wall	0–3
Granularity of the colon wall	0–3
Changes in the vascularity	0–3
Stool consistency	0–3
Mucus production	0–3

**Table 3 ijms-24-16031-t003:** Histological scoring system for oxazolone-induced colitis.

Score	Histological Changes in Oxazolone-Induced Colitis
0	The absence of inflammation
1	Low level of inflammation, with scattered infiltrating mononuclear cells (1–2 foci)
2	Moderate inflammation, with multiple foci
3	High level of inflammation, with increased vascular density and marked wall thickening
4	Maximal severity of inflammation, with transmural leukocyte infiltration and loss of goblet cells

**Table 4 ijms-24-16031-t004:** Primers for qPCR.

Gene Name	Sequence Primer 5′-3′	
*Il1β*	Forward Reverse	GCCAGTGAAATGATGGCTTATT AGGAGCACTTCATCTGTTTAGG
*Il6*	Forward Reverse	CACTGGTCTTTTGGAGTTTGAG GGACTTTTGTACTCATCTGCAC
*Il10*	Forward Reverse	CTTGCTGGAGGACTTTAAGGGTTAC CTTGATGTCTGGGTCTTGGTTCTC
*Tnf-α*	Forward Reverse	CGCCTTGGATTGACAAAC CCTTCCGTGTTCCTACCC
*18s rRNA*	Forward Reverse	GGCGGAGTCCTAAGAGCAAC GGCGGAGTCCTAAGAGCAAC

## Data Availability

Data are contained within the article.

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
