# Peer review of "Effect of a Flavonoid Combination of Apigenin and Epigallocatechin-3-Gallate on Alleviating Intestinal Inflammation in Experimental Colitis Models"

_ijms, 2023, doi:10.3390/ijms242216031_

Round 1
Reviewer 1 Report
Comments and Suggestions for Authors
I have several major comments that need to be addressed:
INTRODUCTION
1. The Introduction provides a general overview of IBD, its types, and its increasing prevalence; however, it lacks a clear focus on the specific research question or problem the study aims to address. It is important to clearly state the gap in the current research that this study intends to fill. Specify the research question or hypothesis to provide readers with a clear direction for the study.
2. While the Introduction mentions several flavonoids and their potential effects on IBD, it lacks a comprehensive review of the existing literature. Include a brief but critical review of previous studies on flavonoids and IBD. Discuss key findings and controversies in the field, highlighting the gaps in knowledge that the current study intends to address. This will provide context and rationale for your research.
3. The Flavo-Natin has been introduced without providing a strong justification for its selection. Why was Flavo-Natin chosen among various other flavonoid combinations? What evidence or rationale supports its selection for this study? Provide a brief background on Flavo-Natin, explaining why it is a suitable candidate for investigation. This will enhance the rationale behind your experimental approach.
4. The Introduction briefly mentions the use of a hapten-mediated experimental colitis model but lacks details on the experimental design. Provide a concise description of the experimental methods and rationale for choosing this particular model. Explain why this model is suitable for studying the protective effects of flavonoids. Additionally, mention any specific parameters or outcomes you will be measuring to assess the protective effect.
METHODS
1. The methods section provides a detailed account of experimental procedures, but some steps could be further clarified. For instance, in the TEER measurement section, provide specific details about the equipment used, electrode calibration, and the units of measurement for TEER. This will enhance the reproducibility of the experiment for readers attempting to replicate it.
2. It is crucial to provide information on how you maintained consistency in experimental conditions, particularly in cell culture and animal studies. Details such as temperature, humidity, and lighting conditions in the laboratory should be mentioned. Additionally, specify how you controlled variables like the age, sex, and genetic background of the mice to ensure the reliability of your results.
3. Clearly state how potential biases and confounding variables were controlled. Explain the randomization process for allocating mice to different groups in animal studies. Additionally, discuss how the researchers were blinded to treatment groups during assessments to minimize observational bias.
4. Provide information on the validation steps undertaken for techniques such as immunofluorescence staining and qPCR. Include positive and negative controls where applicable to demonstrate the reliability of the results. Mention if the experiments were repeated independently to confirm the findings and enhance the robustness of the study.
5. The statistical methods are mentioned briefly. Ensure that the statistical methods are justified, consider adjustments for multiple comparisons, include effect sizes where applicable, and address assumptions to enhance the overall statistical rigor of the study.
DISCUSSION
1. The discussion provides a detailed account of the experimental outcomes but lacks in-depth interpretation. For instance, when discussing the discrepancy between the effect of individual flavonoids (such as apigenin) and Flavo-Natin, delve into possible molecular mechanisms. Why does apigenin decrease TEER while Flavo-Natin, containing apigenin, increases it? Speculate on the interactions and pathways that might be at play.
2. While you briefly mention studies by Watson et al., Fu et al., and Wu et al., there's a need for a more detailed comparison. Compare your findings, especially the dose-dependent effects of Flavo-Natin, with these studies. Discuss the differences in experimental setups, dosages, and administration methods to provide context for your results.
3. The discussion about alterations in the intestinal microbiota is interesting. However, it's currently quite general. Provide more context on the role of Alloprevotella and Akkermansia in intestinal health. Discuss how changes in these specific microbial populations might influence the observed protective effects.
4. Discuss the limitations of your study, such as the small sample size for microbiota analysis and the use of a specific animal model. Address the potential impact these limitations might have on the interpretation of results. Additionally, propose future directions for research. What unanswered questions has this study raised? What further experiments or clinical studies could validate or expand upon your findings?
Minor editing of English language required.
Author Response
Dear reviewer,
Thanks for your valuable comments. I have revised all points based on your advice. Please see the following:
INTRODUCTION
- The Introduction provides a general overview of IBD, its types, and its increasing prevalence; however, it lacks a clear focus on the specific research question or problem the study aims to address. It is important to clearly state the gap in the current research that this study intends to fill. Specify the research question or hypothesis to provide readers with a clear direction for the study.
Answer: Thanks for your precious advice. Line 37-48, added the background of the study aims.
- While the Introduction mentions several flavonoids and their potential effects on IBD, it lacks a comprehensive review of the existing literature. Include a brief but critical review of previous studies on flavonoids and IBD. Discuss key findings and controversies in the field, highlighting the gaps in knowledge that the current study intends to address. This will provide context and rationale for your research.
Answer: Thanks for your precious advice. Line 50-68, added the comprehensive review of the existing literature and critical review of previous studies on flavonoid and IBD.
- The Flavo-Natin has been introduced without providing a strong justification for its selection. Why was Flavo-Natin chosen among various other flavonoid combinations? What evidence or rationale supports its selection for this study? Provide a brief background on Flavo-Natin, explaining why it is a suitable candidate for investigation. This will enhance the rationale behind your experimental approach.
Answer: Thanks for your precious advice. Line 70-77, Provided a brief background on Flavo-Natin, explained why it is a suitable candidate for investigation.
- The Introduction briefly mentions the use of a hapten-mediated experimental colitis model but lacks details on the experimental design. Provide a concise description of the experimental methods and rationale for choosing this particular model. Explain why this model is suitable for studying the protective effects of flavonoids. Additionally, mention any specific parameters or outcomes you will be measuring to assess the protective effect.
Answer: Thanks for your precious advice. Line 79-90, explained why this model is suitable for studying the protective effects of flavonoids and mentioned some specific parameters measured to assess the protective effect.
METHODS
- The methods section provides a detailed account of experimental procedures, but some steps could be further clarified. For instance, in the TEER measurement section, provide specific details about the equipment used, electrode calibration, and the units of measurement for TEER. This will enhance the reproducibility of the experiment for readers attempting to replicate it.
Answer: Thanks for your precious advice. Line 357-367, provided specific details about the equipment used, electrode calibration, and the units of measurement for TEER.
- It is crucial to provide information on how you maintained consistency in experimental conditions, particularly in cell culture and animal studies. Details such as temperature, humidity, and lighting conditions in the laboratory should be mentioned. Additionally, specify how you controlled variables like the age, sex, and genetic background of the mice to ensure the reliability of your results.
Answer: Thanks for your precious advice. Line 405-413, added temperature, humidity, and lighting conditions in the animal facility and the age, sex, and genetic background of the mice.
- Clearly state how potential biases and confounding variables were controlled. Explain the randomization process for allocating mice to different groups in animal studies. Additionally, discuss how the researchers were blinded to treatment groups during assessments to minimize observational bias.
Answer: Thanks for your precious advice. Line 444-461, explained the randomization process for allocating mice to different groups and discussed how the researchers were blinded to treatment groups during assessments to minimize observational bias.
- Provide information on the validation steps undertaken for techniques such as immunofluorescence staining and qPCR. Include positive and negative controls where applicable to demonstrate the reliability of the results. Mention if the experiments were repeated independently to confirm the findings and enhance the robustness of the study.
Answer: Thanks for your precious advice. Line 516-520, added positive and negative controls where applicable to demonstrate the reliability of the immunofluorescence staining. Line 530-531, added qPCR were repeated independently to confirm the results.
- The statistical methods are mentioned briefly. Ensure that the statistical methods are justified, consider adjustments for multiple comparisons, include effect sizes where applicable, and address assumptions to enhance the overall statistical rigor of the study.
Answer: Thanks for your precious advice. Line 556-564, rewrite the statistical methods.
DISCUSSION
- The discussion provides a detailed account of the experimental outcomes but lacks in-depth interpretation. For instance, when discussing the discrepancy between the effect of individual flavonoids (such as apigenin) and Flavo-Natin, delve into possible molecular mechanisms. Why does apigenin decrease TEER while Flavo-Natin, containing apigenin, increases it? Speculate on the interactions and pathways that might be at play.
Answer: Thanks for your precious advice. Line 265-273, explained why apigenin decreases TEER while Flavo-Natin, containing apigenin, increases it.
- While you briefly mention studies by Watson et al., Fu et al., and Wu et al., there's a need for a more detailed comparison. Compare your findings, especially the dose-dependent effects of Flavo-Natin, with these studies. Discuss the differences in experimental setups, dosages, and administration methods to provide context for your results.
Answer: Thanks for your precious advice. Line 325-333, compared the dose-dependent effects of Flavo-Natin, with other published studies and discussed the differences in experimental setups, dosages, and administration methods.
- The discussion about alterations in the intestinal microbiota is interesting. However, it's currently quite general. Provide more context on the role of Alloprevotella and Akkermansia in intestinal health. Discuss how changes in these specific microbial populations might influence the observed protective effects.
Answer: Thanks for your precious advice. Line 294-308, discussed how changes in these specific microbial populations might influence the observed protective effects.
- Discuss the limitations of your study, such as the small sample size for microbiota analysis and the use of a specific animal model. Address the potential impact these limitations might have on the interpretation of results. Additionally, propose future directions for research. What unanswered questions has this study raised? What further experiments or clinical studies could validate or expand upon your findings?
Answer: Thanks for your precious advice. Line 338-346, discussed the limitations of our study.
Best wishes,
Mingrui Li
Reviewer 2 Report
Comments and Suggestions for Authors
This is a well-written original study regarding the effect of different flavonoids in Inflammatory Bowel Disease.
Some minor changes are required
1) The title should be more informative regarding the type of study, the intervention etc. for the readers. In the current form, it is more suitable for a review article and not for an original study. Please change it according to PICO (evidence based medicine-patient/population, intervention, comparison, outcomes).
2) The Introduction section should be at least 2 large paragraphs, providing more in depth information regarding the topic of investigation, and more importantly the aim of the present study. In this manuscript, the aim of the study is not well-written, please rephrase and provide more info regarding flavanoids and their relationship with IBD.
Comments on the Quality of English LanguageMinor English language grammar and syntax editing is required.
Author Response
Dear reviewer,
Thanks for your valuable comments. I have revised all points based on your advice. Please see the following:
- The title should be more informative regarding the type of study, the intervention etc. for the readers. In the current form, it is more suitable for a review article and not for an original study. Please change it according to PICO (evidence based medicine-patient/population, intervention, comparison, outcomes).
Answer: Thanks for your precious advice. The title has been changed.
- The Introduction section should be at least 2 large paragraphs, providing more in-depth information regarding the topic of investigation, and more importantly the aim of the present study. In this manuscript, the aim of the study is not well-written, please rephrase and provide more info regarding flavanoids and their relationship with IBD.
Answer: Thanks for your precious advice. In introduction part, rephrased the aim of the study and provided more info regarding flavonoids and their relationship with IBD.
Best wishes,
Mingrui Li
Round 2
Reviewer 1 Report
Comments and Suggestions for Authors
Thanks to authors for the revisions.